# Hereditary variants of unknown significance in African American women with breast cancer

**J. Tyson McDonald**[1‡], **Luisel J. Ricks-Santi**[2,3‡]*

**1** Department of Radiation Medicine, Georgetown University School of Medicine, Washington, DC, United States of America, **2** Cancer Research Center, Hampton University, Hampton, VA, United States of America, **3** Department of Pharmacotherapy and Translational Research, College of Medicine, University of Florida, Gainesville, FL, United States of America

‡ JTM and LJR are joint senior authors on this work.
* lrickssanti@cop.ufl.edu

**Data Availability Statement:** All relevant data are within the paper and its Supporting Information files. Exome data were also deposited in dbGAP (phs002977).

**Funding:** This work was supported by the Division of Cancer Epidemiology and Genetics, National

## Abstract

Expanded implementation of genetic sequencing has precipitously increased the discovery of germline and somatic variants. The direct benefit of identifying variants in actionable genes may lead to risk reduction strategies such as increased surveillance, prophylactic surgery, as well as lifestyle modifications to reduce morbidity and mortality. However, patients with African ancestry are more likely to receive inconclusive genetic testing results due to an increased number of variants of unknown significance decreasing the utility and impact on disease management and prevention. This study examines whole exome sequencing results from germline DNA samples in African American women with a family history of cancer including 37 cases that were diagnosed with breast cancer and 51 family members. Self-identified ancestry was validated and compared to the 1000 genomes population. The analysis of sequencing results was limited to 85 genes from three clinically available common genetic screening platforms. This target region had a total of 993 variants of which 6 (<1%) were pathogenic or likely pathogenic, 736 (74.1%) were benign, and 170 (17.1%) were classified as a variant of unknown significance. There was an average of 3.4±1.8 variants with an unknown significance per individual and 85 of 88 individuals (96.6%) harbored at least one of these in the targeted genes. Pathogenic or likely pathogenic variants were only found in 6 individuals for the *BRCA1* (p.R1726fs, rs80357867), *BRCA2* (p.K589fs, rs397507606 & p.L2805fs, rs397507402), *RAD50* (p.E995fs, rs587780154), *ATM* (p.V2424G, rs28904921), or *MUTYH* (p.G396D, rs36053993) genes. Strategies to functionally validate the remaining variants of unknown significance, especially in understudied and hereditary cancer populations, are greatly needed to increase the clinical utility and utilization of clinical genetic screening platforms to reduce cancer incidence and mortality.

Cancer Institute, R15CA239100, Dr. Luisel J Ricks-Santi, Dr. J. Tyson McDonald.

**Competing interests:** The authors have declared that no competing interests exist.

## Introduction

Familial and hereditary breast cancer (BCa) makes up 20–30% and 5–10% of all breast cancers, respectively [1]. Many "actionable" genes have been discovered, such as *BRCA1* and *BRCA2* which confer the greatest risk of BCa, at 40–87% for *BRCA1* mutation carriers and 18–88% for *BRCA2* mutation carriers [2]. In adults, actionable genes are classified as those with deleterious mutation(s) whose phenotype results in a specific, defined medical recommendation(s). Thus, these genes inform patients and doctors in the proper deployment of risk reduction strategies such as early detection, prophylactic surgery, chemoprevention, lifestyle modifications, and treatment. The functional consequences of these genes must be supported by prior experimental evidence and when considered in treatment must improve an outcome(s) in terms of mortality or the avoidance of significant morbidity [3, 4].

More recently, the clinical use of low-cost genetic sequencing in multiple genes, or gene panels, is increasing [5] due to the discovery of additional actionable genes revealed in the last 20 years. However, actionable genes such as *BRCA1* and *BRCA2* also harbor a myriad of genetic variants of unknown clinical significance (VUSs) hampering prevention and cancer management efforts, especially in African American (AA) women who report a higher frequency of VUS in *BRCA1/2* [6]. Not all genetic variants in actionable genes have clinical relevance and in fact, many variants are benign. Nevertheless, wide scale implementation of genetic testing has resulted in an increased observation of VUSs without documented clinical relevance limiting their clinical utility especially in diverse populations [7–11].

The discoveries of causative functional alleles have far-reaching implications for the early detection of cancer, cancer prevention efforts, and the elimination of cancer disparities. In the case of *BRCA1/2*, and other genes, there are specific guidelines for prevention, management and treatment of breast and ovarian cancer patients with pathogenic, disease-causing mutations. The role of *BRCA1/2* in therapeutic regimens has emerged based on the DNA repair of double stranded breaks. A synthetic strategy for cancer therapy has been developed in which DNA damaging chemotherapeutic agents that cause single-stranded breaks are being used in combination with poly-ADP ribose polymerase (PARP) inhibitors, which inhibit single-stranded DNA repair. This approach has been shown to be particularly effective in BRCA mutation carriers, as these tumors are unable to repair the double-stranded breaks [12–14] and may be the case for other cancer predisposition genes whose roles are overrepresented in DNA repair pathways. For example, PARP inhibitors appear to also have utility, although limited, in metastatic breast cancer and TNBC with *PALB2* mutations and somatic *BRCA1* mutations [15]. Remarkably, there are at least 3 clinical trials testing this therapeutic strategy in non-BRCA mutation carriers with deficient homologous recombinational repair: NCT03344965, NCT03742895 and NCT03367689. Still, VUSs, even in actionable genes, hinders clinical decision making in patients undergoing genetic testing and can lead to significant distress in young and AA women [16]. This study will provide insight into the scope and identification of genetic variants with broad implications in familial BCa patients, especially in AA women.

## Methods

### DNA samples

DNA samples were from the African American Familial Breast Cancer Study (AAFBC) [17–20] and the Breast Cancer Determinants in Women from Washington, DC study. The original study was approved by the Howard University Institutional Review Board (IRB97CC01). This study met the criteria for exemption as a secondary analysis of existing data was performed

and the data were analyzed anonymously and was also approved by the Howard University Institutional Review Board (09-MED-86). These participants were recruited initially from AA women with a primary diagnosis of BCa and a family history of cancer. Additional participants were then recruited from their family members including first-degree, second-degree, and distant relatives in the same lineage. Eligibility for this study included meeting one or more of the following criteria: 1) having multiple cases (≥2) of breast or ovarian cancer in the family; 2) having early onset breast cancer (≤40 years old); 3) having bilateral breast cancer; 4) having breast and ovarian cancer in the same individual, or 5) having male breast cancer in the family. Positive family history was defined as having multiple cases (≥2) of breast or ovarian cancer in the same side of the family. A total of 88 individuals were selected for whole exome sequencing analysis. These individuals represent affected probands (n = 37) and/or their previously affected and/or unaffected family members (n = 51). Eleven cases reported family histories of other cancers including, but not limited to lung, colorectal, and uterine cancer. The mean age for the probands, unaffected family members and affected family members was 53.76±14.25, 48.80±16.45, and 41.2±8.67, respectively. DNA samples were examined by agarose gel electrophoresis to test for DNA degradation and potential contamination. DNA concentration was measured using a Qubit 2.0 Fluorometer (Thermo Fisher Science, Waltham, MA).

## Whole exome sequencing (WES)

The Agilent SureSelect Human All ExonV6 kit (Agilent Technologies, CA) was used to generate sequencing libraries following manufacturer's recommendations. Products were purified using AMPure XP system (Beckman Coulter, Beverly, USA) and quantified using the Agilent high sensitivity DNA assay on the 2100 Bioanalyzer Instrument (Agilent, Santa Clara, CA). Sample libraries were sequenced using Illumina protocols by Novogene (Novogene Co., Ltd. Durham, NC). Data has been submitted to dbGAP (phs002977).

## WES data processing, genomic copy number analysis, and variant calling

Following quality control analysis, raw data fastq files were processed according to the Genome Analysis Toolkit's (GATK) Best Practices workflow [21–23]. Briefly, reads were aligned to the hg38 reference genome using the Burrows-Wheeler Aligner v0.7.17 [24, 25] followed by duplicate removal, base quality score recalibration, haplotype calling for variant identification using the joint calling process, and filtered using the variant quality score recalibration from GATK v4.1.4.1. Variants were limited to the target regions used for library preparation and the final set of variants were annotated using SNPeff with information from the NCBI's ClinVar and dbSNP databases. Oncoplots and heatmaps were generated with maftools v2.2.10 [26] and pheatmap v1.0.12 respectively. Genomic copy number evaluation was performed with ExomeDepth [27]. Germline variants in the same target regions were also identified from AA (N = 174) or CA (N = 348) samples in the Cancer Genome Atlas Breast Invasive Carcinoma (TCGA-BRCA) data collection.

## Estimation of genetic ancestry

The study population was compared individuals with known ancestry from the 1000 Genomes Project dataset. Genotypes used were from the data release of phased biallelic SNP plus INDEL call sets for unrelated samples aligned to GRCh38 (version 3-12-2019 [28]). There were 2,548 samples from 26 populations representing the 5 superpopulations (AFR, African; AMR, Ad Mixed American; EAS, East Asian; EUR, European; SAS, South Asian). There were 10,512 exon coding variants in the current dataset that matched to the 1000 Genomes dataset. Following filtering with PLINK v1.9 [29] for variants with a >10% minor allele frequency using a

50kb window that excluded variants in linkage, there were 645 variants used for principal component analysis (PCA). PCA was performed on the study population and 26 populations from the 1000 Genomes project by plotting the principle eigenvalue components to visualize clustering of similar ancestries [30, 31]. Population ancestry admixture was performed on the study population and 4 populations from the 1000 Genomes dataset (CEU, ASW, YRI, and CHB) using the Bayesian clustering approach of the STRUCTURE v2.3.4 program [32]. From the previous set of 645 variants used in the PCA analysis, 579 variants were single-nucleotide variants that were used for admixture analysis. The estimation value for the number of populations (K) was set from 3 to 5 using a 10,000 burn-in followed by 20,000 iterations with 2 replicates each. There results were combined and plotted using the software package Clumpak [33].

### Statistics

Statistical differences in the total number of variants with a pathogenic clinical significance were made using an independent t-test. P-values less than 0.05 were considered significant.

## Results

### Estimation of individual genetic ancestry

DNA samples were previously collected from self-identified African American women with a primary diagnosis of BCa and a family history of cancer. Additional samples were collected from family members. A total of 88 DNA samples comprising of 37 probands and 51 family members were successfully analyzed using whole exome sequencing. Following data processing using the Genome Analysis Toolkit (GATK) Best Practices workflow [21–23], the subsequent variants were filtered to include exon targeted regions following annotation with current dbSNP and ClinVar databases.

Estimation of individual ancestry was performed using appropriate variants present in both the study population and the 1000 Genomes project database. First, the resulting variants were used to perform a principal component analysis (PCA). As expected, the self-identified AA women clustered closely with populations of African ancestry (Fig 1A–1C). Second, the study population was compared to four populations (CHB, CEU, YRI, and ASW) using the admixture model from STRUCTURE to estimate genetic ancestry. Examining the results using a bar plot or ternary plot demonstrated that the study population displayed similar admixture to the ASW population (Fig 1D and 1E). Thus, this population of women with African ancestry, BCa, and a familial history of cancer represents a unique dataset to study germline genetic variants that may predispose individuals to cancer.

### Variants within clinical cancer screening panel

While somatic and germline WES is gaining clinical utility, the use of targeted sequencing panels are widely available especially in the case of suspected hereditary cancers [34–38]. There were 85 unique genes targeting actionable variants in largely DNA-repair genes that were present on three commercially available clinical screening platforms from the Invitae Corporation [39], Ambry Genetics [40], and Myriad Genetics [41] (Fig 2A). There were 28 genes (33%) in common with all 3 panels while 23 gene targets (27.1%) were unique to only one panel. Within these 85 targeted gene regions, there were 959 SNPs and 34 INDELs from the cohort of 88 AA women. These variants were examined for ClinVar annotated clinical significance as benign/likely benign, pathogenic/likely pathogenic, uncertain significance, or conflicting interpretations of pathogenicity (Fig 2B). While 74.1% of the total variants were classified as benign or likely benign, there were 170 VUSs (17.1%) with an average of 3.4±1.8 VUSs per individual

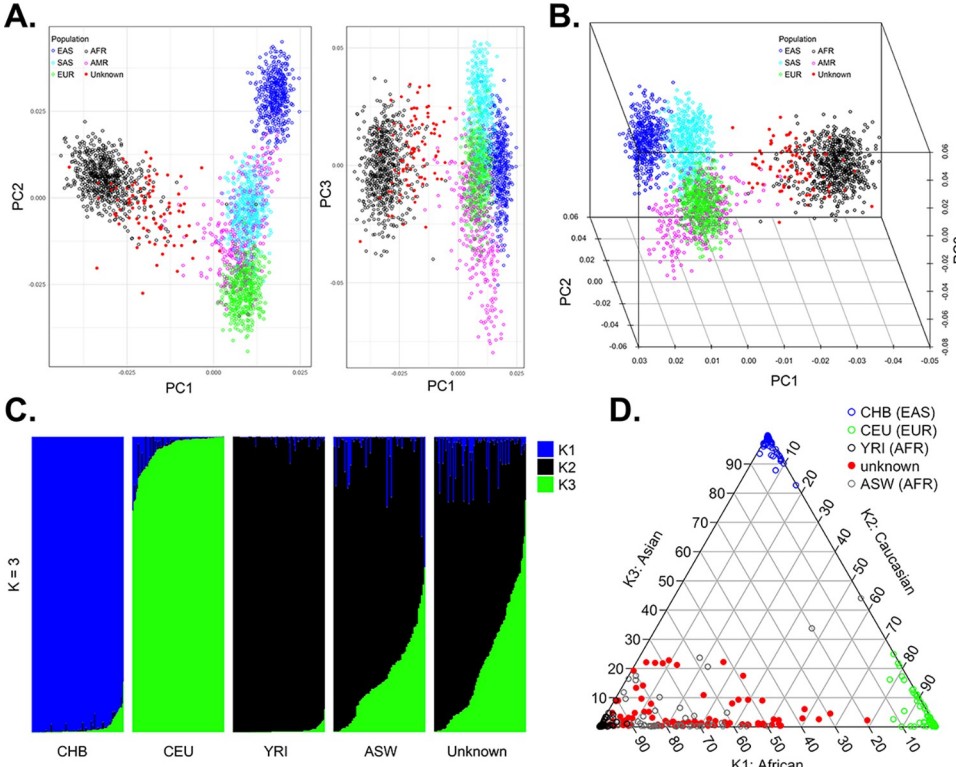

**Fig 1. Estimation of individual genetic ancestry.** (A-C) Principal component analysis (PCA) was used to visualize individual clustering compared to the 1000 Genomes Project 5 superpopulations. (D-E) STRUCTURE's admixture model based on three ancestral populations (K = 3) was used to compare the study population to CEU, YRI, CHB, and ASW populations. The data is displayed using a (D) bar or (E) ternary plot.

(Fig 2C, S1 and S2 Tables). Detectable changes in genomic copy number were found in only 9 of 993 variants (0.01%) and only 2 of these variants were classified as a VUS (S2 Table). Notably, 85 of 88 individuals (96.6%) had at least one germline VUS within these 85 clinically targeted genes.

To better understand and prioritize the VUS in this cohort, 9 *in silico* prediction algorithms were assembled. A consensus score was tallied for the most damaging prediction in each algorithm (Fig 3). The *CHEK2* variant (H143R, rs587782300) had the highest consensus score that was predicted to be damaging in 8 of the 9 algorithms. This variant was first identified in women with hereditary ovarian cancer [42] and subsequently in AA women with invasive BCa [43] as well as those suspected of Lynch syndrome [44]. This variant is located in the forkhead-associated (FHA) functional domain of *CHEK2* and was found to impair DNA repair activity in yeast [42, 45]. However, functional validation for the H143R variant in human models is lacking. The next highest consensus score predicted to be damaging in 7 of the 9 algorithms included variants in the *CDKN1B* (P35L, rs375297371) and *TSC2* (R1751H, rs373365980 & D907Y, rs1287273870) genes. Functional studies in human models and epidemiological results in larger cancer cohorts are lacking for these VUSs. In mutant yeast cells, the *CDKN1B* variant did not affect binding to the VHL protein or affect the yeast growth rates [46] and the *TSC2* variants have not observed in published studies for its associated diseases. Thus, the identification in this cohort and *in silico* scoring results may assist to prioritize these VUSs for functional evaluation in experimental human models.

There is no gold standard to classify VUSs using *in silico* results which may overlook unpredicted consequences. For instance, the clinical significance for three VUSs with conflicting

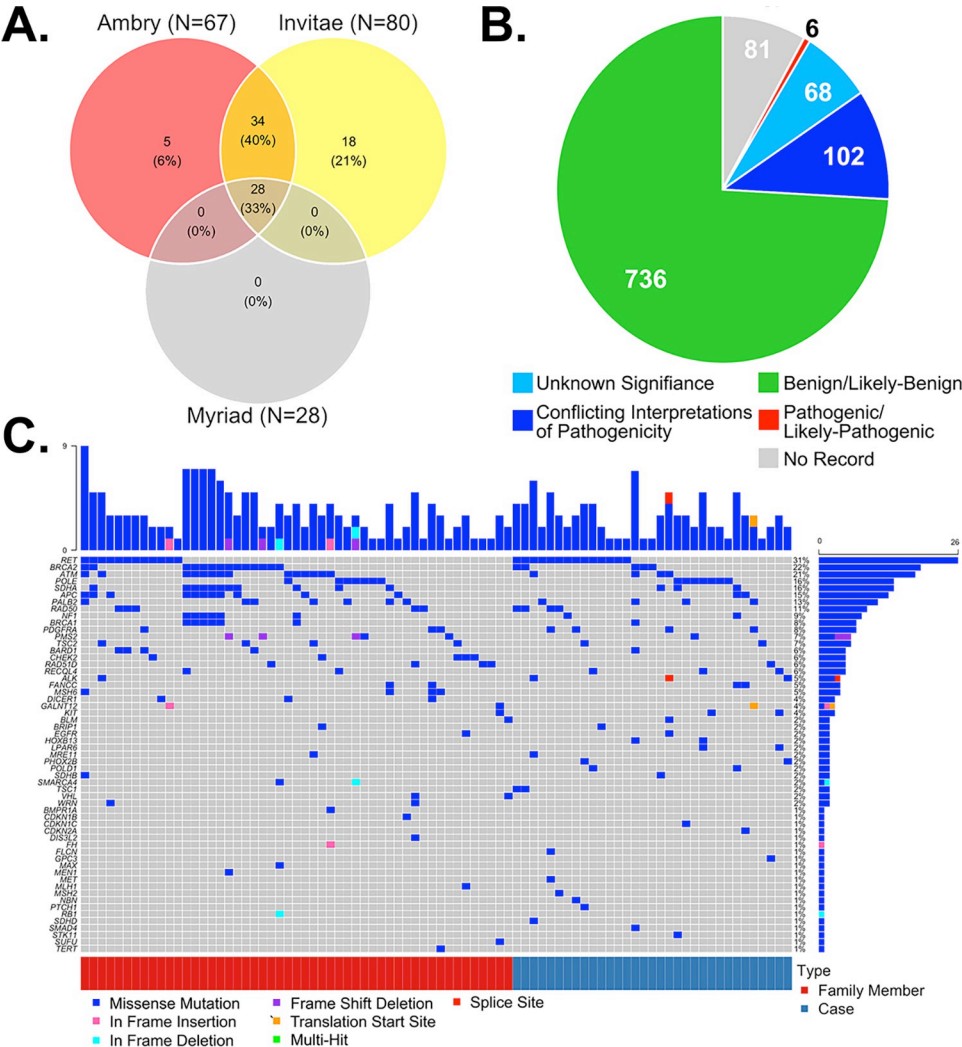

**Fig 2. Variants in clinical cancer screening panel genes.** (A) Overlap of Invitae, Ambry, and Myriad breast cancer/cancer panels resulting in 85 clinically targeted genes. (B) Classification of clinical significance in 959 SNPs and 34 INDELs in the 85 actionable genes from common clinical cancer screening panels. (C) Oncoplot of genes with one or more germline VUS found in 69 of 88 individuals.

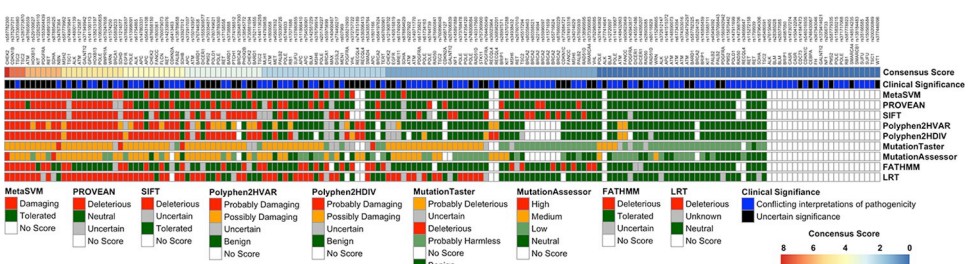

**Fig 3. *In silico* prediction scores for VUSs.** There were 170 VUSs with a clinical significance annotated as unknown (black) or with conflicting interpretation of pathogenicity (dark blue). A consensus score (top bar) was generated by totaling the number predictive algorithms with a deleterious/damaging or equivalent score (red).

**Table 1. Pathogenic or likely pathogenic variants from the targeted clinical cancer screening genes.**

| Gene | RSID | REF | ALT | Consequence | Proband or Family Member | Clinical Significance |
|------|------|-----|-----|-------------|--------------------------|------------------------|
| *HOXB13* | rs138213197 | C | T | G84E; Missense | Proband | Conflicting interpretation of pathogenicity |
| *FH* | rs367543046 | A | A**TTT** | N478KN; In-frame Insertion | Family Member | |
| *CHEK2* | rs121908702 | G | A | E239K, Missense | Family Member | |
| *ATM* | rs28904921 | T | G | V2424G; Missense | Proband | Likely Pathogenic/ Pathogenic |
| *MUTYH* | rs36053993 | C | T | G396D; Missense | Family Member | |
| *BRCA1* | rs80357867 | T**CTTT**C | TC | R1726fs; Frameshift | Proband | Pathogenic |
| *BRCA2* | rs397507606 | T**AA** | T | K589fs; Frameshift | Family Member | |
| *BRCA2* | rs397507402 | T**TA** | T | L2805fs; Frameshift | Proband | |
| *RAD50* | rs587780154 | C**AAAG** | C | E995fs; Frameshift | Family Member | |

interpretations of pathogenicity also had low consensus scores but epidemiolocal studies indicated a pathogenic or likely pathogenic status (Table 1). Reports have linked the *HOXB13* germline variant (G84E, rs138213197) to an increased risk of prostate cancer in men of European descent [47, 48] and also found an increase in overall cancer risk in a pooled analysis of 25 epidemiological studies with 145,257 participates [49]. Though, there is also inconclusive evidence for prostate cancer reported for this variant. The in-frame insertion in the *FH* gene (N478KN, rs367543046) was recently found in 24 of individuals with various cancers and 11 individuals with renal cell carcinoma from a pool of 7,571 patients [50], however a significant association of the variant to cancer was not found. Lastly, the *CHEK2* gene variant (E239K, rs121908702) was initially identified in men with prostate cancer [51] but has also had conflicting association with cases [52] and controls [53] in two separate breast cancer cohorts. Interestingly, *in vitro* studies did result in an intermediate reduction of *CHEK2* kinase activity [53]. While these three variants are classified with uncertain significance or conflicting interpretations of pathogenicity, it may be necessary to examine a larger cohort of African American women with cancer to confirm a deleterious association specifically in this population.

There were only six variants with a likely pathogenic or pathogenic clinical significance that were uncovered in our hereditary cohort (Table 1). The *ATM* variant (rs28904921) was recently compared to other pathogenic missense or truncating variants in the *ATM* gene in a large cohort of 627,742 hereditary cancer patients and found a higher increased risk of invasive ductal BCa (OR 3.76, 95% CI 2.76–5.21) for carriers of this variant [54]. The *MUTYH* gene variant (rs36053993) has been well described as a pathogenic variant in patients causing adenomatous polyposis and colorectal cancer [55–57] through a reduction of DNA binding and glycosylase activity [58, 59]. There were two pathogenic variants identified in the *BRCA2* gene and one in each of the *BRCA1* and *RAD50* genes all of which resulted in a frameshift and truncated protein. The *BRCA1* variant (rs80357867) was originally identified in AA families with early onset BCa [60]. Since then, a number of studies have since identified this variant in American individuals [43, 61, 62] as well as those that are inhabitants of Slovenia [63], Brazil [64], Trinidad and Tobago [65], and Germany [66]. More recently, this variant was also identified in a cohort of 615 pediatric Rhabdomyosarcoma patients [67]. Both *BRCA2* variants (rs397507606 and rs397507402) were originally identified in a study of AA women with familial and early onset cases of breast and/or ovarian cancer [18]. The RAD50 frameshift variant (rs587780154) has been linked to familial BCa and Nijimegen breakage syndrome [68, 69].

Finally, a comparison to the number of pathogenic variants in these 85 clinically targeted genes found in AA and CA women in the Cancer Genome Atlas (TCGA) Breast Invasive Carcinoma (BRCA) dataset was performed. There were no significance differences in the number of pathogenic variants found in the TCGA AA population and our hereditary cohort.

However, there were approximately 3 times as many pathogenic variants identified in the TCGA individuals with CA ancestry (N = 348) compared to TCGA AA ancestry (N = 174; p-value = 0.0004) or individuals examined in this study (N = 88; p-value = 0.003). Although evaluation of the six damaging variants revealed significant links to an increase in cancer risk, this comparison demonstrates a potentially limited utility of targeted multi-gene panels in individuals with AA ancestry.

## Discussion

There is a striking lack of diversity as well as an underreporting of race as a demographic variable in genomic sequencing studies [70]. Of the published genome-wide association studies to identify genes associated with predisposition to BCa, few included women of African descent and even fewer have been done in AA families [17, 43, 71]. Therefore, the objective of this study was characterize genetic variants in 85 clinically actionable genes in a high-risk AA familial BCa cohort. Of the variants identified in those genes, the majority (74.1%) were benign, while 17.1% were VUSs or had conflicting pathogenicity. Each individual averaged approximately 3.4 (±1.8) VUSs, and all but 3 individuals had at least 1 VUS. Remarkably, only 4 pathogenic and 2 likely pathogenic variants were identified in our familial BCa cohort demonstrating a significant need to improve the clinical utility of genetic/genomic testing in diverse populations.

Compared to Caucasian (CA) women, AA women with BCa are often diagnosed at a younger age, have higher grade tumors, have a higher frequency of triple negative BCa, and have tumors that are more lethal with less successful treatment options (reviewed by Newman *et al.* [72, 73]). These are characteristics consistent with genetic predisposition, thus identifying the genes that predispose AA women to BCa could improve outcomes in a subset of AA BCa cases. Additionally, the identification of actionable genetic variants predisposing more than 5–10% of all BCa cases also guides the management and treatment of cancer in those patients testing positive for pathogenic germline or somatic variants in any of the panel of 85 actionable genes examined in this study. Because the majority of genes found in these panels are within DNA repair pathways, therapeutics that exploit DNA repair deficiencies, such as PARP inhibitors, have become successful strategies for patients diagnosed with hereditary breast and ovarian cancer. A recent trial of one PARP inhibitor demonstrated partial response or stable disease in breast, ovarian and prostate cancer subjects [74]. Importantly, triple negative BCa is often associated with mutations in DNA repair genes such as *BRCA1*, making the therapeutic targeting of the pathway a viable option for the difficult to treat tumor type [15].

Wide scale implementation of genetic counseling has caused the exponential growth of clinical genetic testing and unintended discoveries of germline variants and VUSs without documented clinical relevance resulting in inconclusive findings. Within actionable genes are a myriad of variants whose function remains unknown. Although VUSs exist for all populations, the prevalence of VUS is significantly higher in patients with African ancestry. Sequencing results from a multi-center study in Florida, Arizona, New York and Wisconsin using the Invitae Multi-Cancer panel on 2,571 CAs and 110 AAs with solid tumors observed similar pathogenic variants (13.3% in CAs; 12.7% in AAs) while VUSs were substantially higher in AAs (46.1% in CAs; 65.5% in AAs) [75]. Tatieni *et al.* [11] and Roberts *et al.* [10] also found a reduced likelihood of pathogenic variants in AAs compared to non-Hispanic whites while both studies confirmed the higher frequency of VUS in AAs. In a smaller study, Bishop et al. also demonstrated that AA breast cases were more than 2.5 times more likely to carry at least 1 VUS (40% in AAs vs. 14.5% in CAs). Conversely, using publicly available data, Ndugga-Kabuye *et al.* [8] demonstrated that AA patients with a family history of cancer had a slightly

higher frequency of pathogenic and likely pathogenic variants compared to CAs. Our findings provide further support that AAs with family histories of cancer will present with a lower identification of pathogenic variants along with a high number of clinical VUSs, thus limiting the clinical utility of cancer genomic panels in the group. While there is significant utility for the use of genetics and genomics to identify women at risk for BCa that may benefit from early detection and prevention strategies, AAs will have a reduced benefit from genomic cancer panel testing due to the high number of VUSs. Additionally, VUSs present significant challenges in the counseling of patients and their families regarding the appropriate management for cancer risks. The limited utility may be a result of reduced inclusion of data from diverse populations and the limit in tools that allow the timely functionalization of variants of unknown clinical significance.

Attempts to classify VUSs have been limited. Recently, 138,342 eligible individuals in the Ambry Genetics patient database were examined for *BRCA1/2* variant reclassification based on personal and family history of cancer [76]. The results from 2,383 *BRCA1/2* variants indicated 45 variants were in favor of pathogenicity while 150 were against. However, only 177 (7.4%) of these total variants were observed in 5 or more individual probands. It is notable, however, that the American College of Medical Genetics and Genomics (ACMG), Association for Molecular Pathology, American Society of Clinical Oncology, and College of American Pathologists have provided guidance and recommendations for the classification of variants [77, 78]. ACMG's 5-tier system ranging from benign to pathogenic, evaluates the following criteria: population data, computational and predictive data, functional data, segregation data, de novo data, allelic data, other database and other data. The latter organizations, combined benign and likely benign variants resulting in a similar 4-tier system. Using these guidelines, several studies have reclassified *BRCA1* and *BRCA2* variants in Asian populations [79–81]. Given that that a positive family history increases the likelihood of variant reclassification, our goal is to use the ACMG guidelines to reclassify variants at high frequency in our high-risk BCa patients as reclassifications have a high potential for clinical impact often altering the clinical management of patients [82]. For example, Turner *et al.* showed that with the reclassification of 142 variants, clinical impact changed for 11.3% of those patients [83]. For many of those patients, variant reclassification resulted in an upgrade in reporting category (*i.e.* VUS to likely pathogenic) and altered clinical management, such as recommendations for organ surveillance, survey or cascade testing which tests family members for resulting variants.

Well established *in vitro* or *in vivo* functional studies demonstrating a damaging effect on protein function also provide evidence for support of pathogenicity that meets the ACMG guidelines. Functional characterization of variants in *BRCA1* and *BRCA2* is an established area and assays have been performed to test the impact of variants on ubiquitin ligase activity, transcription activation, protease sensitivity assays, and PARP inhibitor sensitivity assays as readouts (reviewed by Monteiro *et al.* [84]). More recently, *PALB2* has been the subject of functional characterization given its more recent association with hereditary breast and ovarian cancer after its incorporation in early BCa genomic panels. Boonen *et al.* [85] and Wiltshire *et al.* [86] have developed a high-throughput *in vitro* CRISPR and reporter protocols that allow for the simultaneous screen of multiple VUSs using PARP inhibitor cytotoxic assays, DNA repair assays, and G2/M cell cycle checkpoint assays. CRISPR-based assays have more recently be utilized for the functional reclassification of *BRCA1* [87]. Finally, a recent pan-cancer analysis of clinically actionable oncogenic driver genes in the Cancer Genome Atlas (TCGA) was performed [88]. From over 9,000 tumor samples in 33 cancer types, they found 57% of tumors harbor actionable variants and were able to functionally validate 39 of 46 variants using two cell lines in survival and growth assays for carcinogenesis. The use of functional assays provide a promising step forward in identifying and verifying variants influencing tumor progression.

## Conclusions

The limited association of VUSs as well as time consuming and limited functional testing of VUSs with genetically defined cancer syndromes in large next generation sequencing data requires an innovative and high-throughput functional genomics approach. The successful identification and evaluation of VUSs may improve the deployment of personalized cancer prevention and treatment options for women with BCa. With an improvement in the classification of actionable variants, a greater understanding of the risk profile in all populations may be better understood and result in a novel intervention targeted approach to improve patient outcomes.

## Supporting information

**S1 Table. Variants with an unknown significance or conflicting interpretation of clinical significance.** VUSs (N = 170) from the targeted regions of 85 genes with a tallied consensus score for 9 *in silico* prediction algorithms. The review status definition was provided by ClinVar as follows: (3) reviewed by expert panel, (2) two or more submitters with assertion criteria and evidence provided the same interpretation, or (1) multiple submitters provided assertion criteria and evidence but there are conflicting interpretations.
(XLSX)

**S2 Table. Allele frequencies for all the variants found in the targeted regions of 85 genes (N = 993).** The total and population specific allele frequencies (AF) for the current study (N = 88), 1000 Genomes Project, NHLBI GO Exome Sequencing Project (ESP), and Genome Aggregation Database (gnomAD) are shown. Genomic copy number analysis (gCNV) was calculated using ExomeDepth.
(XLSX)

## Acknowledgments

The results published here are in part based upon data generated by the TCGA Research Network: https://www.cancer.gov/tcga.

## Author Contributions

**Conceptualization:** J. Tyson McDonald, Luisel J. Ricks-Santi.

**Data curation:** J. Tyson McDonald.

**Formal analysis:** J. Tyson McDonald.

**Funding acquisition:** J. Tyson McDonald, Luisel J. Ricks-Santi.

**Investigation:** J. Tyson McDonald, Luisel J. Ricks-Santi.

**Methodology:** J. Tyson McDonald, Luisel J. Ricks-Santi.

**Project administration:** Luisel J. Ricks-Santi.

**Resources:** Luisel J. Ricks-Santi.

**Software:** J. Tyson McDonald.

**Supervision:** Luisel J. Ricks-Santi.

**Validation:** J. Tyson McDonald.

**Visualization:** J. Tyson McDonald.

**Writing – original draft:** J. Tyson McDonald, Luisel J. Ricks-Santi.

**Writing – review & editing:** J. Tyson McDonald, Luisel J. Ricks-Santi.

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
