## [Decision Letter · Decision Letter 0]

18 Apr 2022

PONE-D-22-08035Hereditary variants of unknown significance in African American women with breast cancerPLOS ONE

Dear Dr. Ricks-Santi,

Thank you for submitting your manuscript to PLOS ONE. After careful consideration, we feel that it has merit but does not fully meet PLOS ONE’s publication criteria as it currently stands. Therefore, we invite you to submit a revised version of the manuscript that addresses the points raised during the review process.

We look forward to receiving your revised manuscript.

Kind regards,

Alvaro Galli

Academic Editor

PLOS ONE

Journal Requirements:

Reviewers' comments:

Reviewer's Responses to Questions

**Comments to the Author**

1. Is the manuscript technically sound, and do the data support the conclusions?

Reviewer #1: Partly

Reviewer #2: Yes

2. Has the statistical analysis been performed appropriately and rigorously? 

Reviewer #1: N/A

Reviewer #2: Yes

3. Have the authors made all data underlying the findings in their manuscript fully available?

Reviewer #1: No

Reviewer #2: Yes

4. Is the manuscript presented in an intelligible fashion and written in standard English?

Reviewer #1: Yes

Reviewer #2: Yes

5. Review Comments to the Author

Reviewer #1: The manuscript by McDonald and Ricks-Santi evaluates targeted (DNA-repair gene) sequencing findings from 37 women with breast cancer and 51 family members from study cohorts of African American women. This is a fairly straightforward study that characterizes the findings of 85 cancer-related genes from an exome study. The results are not surprising with most sequence variants being identified as benign or VUS and a small proportion (6/37; 16% or 6 of 87—not clear from the text) of the women in this study had a pathogenic variant. There was some missing critical information in the manuscript about the cancer diagnosed in the families, characteristics about the family history and probands, how and many of the variants found were shared across family members. The rationale for including genes that are not associated with hereditary breast cancer through large studies is unclear. Points to be addressed are as follows:

Major:

This is called a high-risk cohort by the authors. Why? What is the definition of high-risk used for this study? (e.g. Were all the women with breast cancer diagnosed under the age of 50?) A definition of family history should also be included.

More description on the 37 women with breast cancer should be included such as age of onset and ER/PR/HER2 status of the breast cancer.

More description on the family members sequenced should be included. Were these affected family members, unaffected or a mix? If affected, did they have cancers other than breast? How many of them shared pathogenic/likely pathogenic variants with the 37 women?

It is not clear from the description of the analysis if analysis was able to detect large copy number changes. This should be noted in the results and discussion and if no copy number changes were identified on analysis this should also be included.

Color Genomics has a database in which variants as well as self-reported race are captured. Have any of the VUS in this study been identified in women of African ancestry? Are any of them unique to individuals with African ancestry in women with cancer or health women and if so what is the allele frequency?

For the conflicting variants, have any been assessed by a ClinVar expert panel or have they been evaluated in BRCAexchange? These calls may have more “weight” than earlier lab reports. That information could go into Figure 3.

A supplement table(s) listing all the variants found in the study, not just the VUS, should be included.

A number of genes included (such as HOXB13, FH) are not considered to be breast cancer susceptibility genes and PVs within these genes do not increase breast cancer risk. It is not clear why these would be included (unless the family history has a lot of other cancers). What do the data look like when ONLY breast cancer related susceptibility genes are included? (Path/LP, VUS, Benign/LB).

The exome sequence data for these 85 genes should be made available (e.g. deposited into dbGAP).

Minor:

Gene names should be italicized.

The term “mutation” (and deleterious mutation) should be replaced with “pathogenic variant”.

Line 70. Would read better as “the role of BRCA1/2 in…”

Lines 176-178. The sentence needs editing. “This variant is located in the forkhead-associated (FHA) functional domain of CHEK2 177 and while it has been found to impair DNA repair activity in yeast [38, 41].

Reviewer #2: Disease history (other cancers, age, bilateral, and etc) of probands and cancer types of family members would be presented for explanation of low frequency of pathogenic varinat in this study group. Also it would be need to represent comparable study results for frequencies of PV, VUS, and benign variants

6. PLOS authors have the option to publish the peer review history of their article (what does this mean?). If published, this will include your full peer review and any attached files.

Reviewer #1: No

Reviewer #2: No

---

## [Author Response · Author response to Decision Letter 0]

8 Jul 2022

We would like to thank the reviewers for their comments. Below, we have responded to each of the concerns raised. 

Reviewer #1:There was some missing critical information in the manuscript about the cancer diagnosed in the families, characteristics about the family history and probands, how and many of the variants found were shared across family members. The rationale for including genes that are not associated with hereditary breast cancer through large studies is unclear. Points to be addressed are as follows:

Major:

This is called a high-risk cohort by the authors. Why? What is the definition of high-risk used for this study? (e.g. Were all the women with breast cancer diagnosed under the age of 50?) A definition of family history should also be included.

This family-case-control study was developed at the Howard University Cancer Center, a historically black university, and funded by the National Cancer Institute. 

The following definition of high-risk and family history is now included in the Methods section (lines 112-116):

“Eligibility for this study included meeting one or more of the following criteria: 1) having multiple cases (≥2) of breast or ovarian cancer in the family; 2) having early onset breast cancer (≤40 years old); 3) having bilateral breast cancer; 4) having breast and ovarian cancer in the same individual, or 5) having male breast cancer in the family. Positive family history was defined as having multiple cases (≥2) of breast or ovarian cancer in the same side of the family.” 

More description on the 37 women with breast cancer should be included such as age of onset and ER/PR/HER2 status of the breast cancer.

Unfortunately, additional status such as ER/PR/HER2 is not available for the 37 women with breast cancer. The mean age was available and has been added in the Methods section (lines 141-143). The mean age for the probands, unaffected family members and affected family members was 53.76±14.25, 48.80±16.45, and 41.2±8.67, respectively. 

More description on the family members sequenced should be included. Were these affected family members, unaffected or a mix? 

These individuals represent affected probands (n=37) and/or their affected and/or unaffected family members (n=51). 

If affected, did they have cancers other than breast? 

Eleven cases reported family histories of other cancers including, but not limited to lung, colorectal, and uterine cancer.

How many of them shared pathogenic/likely pathogenic variants with the 37 women? 

Three of the pathogenic/likely-pathogenic variants were found in 3 of the 37 cases. To make this clear, we have added a column in Table 1 stating if the variant was from a proband or family member. 

It is not clear from the description of the analysis if analysis was able to detect large copy number changes. This should be noted in the results and discussion and if no copy number changes were identified on analysis this should also be included. 

Genome copy number variation (gCNV) has now been run and is available in the new S2 Table. There was minimal impact of gCNV on the resulting variants. Only 9 of 993 variants (0.01%) had a detectable change in gCNV in two genes (7 variants in POLE and 2 variants in RECQL4). Only two of these variants with gCNV were classified VUSs and were both in the POLE gene. The remaining 7 variants were classified as benign or likely-benign.

Color Genomics has a database in which variants as well as self-reported race are captured. Have any of the VUS in this study been identified in women of African ancestry? Are any of them unique to individuals with African ancestry in women with cancer or health women and if so what is the allele frequency? 

We added the allele frequencies from standard and open access datasets from the 1000 Genomes Project, NHLBI GO Exome Sequencing Project (ESP), and Genome Aggregation Database (gnomAD). This data is now shown in the new S2 Table. This data includes allele frequencies from individuals with African ancestry as well as other ancestry populations. For the VUSs, there were 19 of 170 (11.2%) that did not have a reported allele frequency in these databases. 

For the conflicting variants, have any been assessed by a ClinVar expert panel or have they been evaluated in BRCAexchange? These calls may have more “weight” than earlier lab reports. That information could go into Figure 3.

There were 12 VUSs in the BRCA1 or BRCA2, however BRCAexchange lists each of these variants as “not yet reviewed”. For the ClinVar review, there were only 2 of 170 variants that were reviewed by an expert panel, both of which had a low consensus score (0 or 1, on a scale up to 9) provided in Figure 3 and S1 Table. Information on the review status provided by ClinVar has now been added to the S1 Table.

A supplement table(s) listing all the variants found in the study, not just the VUS, should be included.

The new S2 Table now includes a list of the 993 variants found in this study.

A number of genes included (such as HOXB13, FH) are not considered to be breast cancer susceptibility genes and PVs within these genes do not increase breast cancer risk. It is not clear why these would be included (unless the family history has a lot of other cancers). What do the data look like when ONLY breast cancer related susceptibility genes are included? (Path/LP, VUS, Benign/LB). 

As the full dataset of variants is now provided in S2 Table, a specific comparison of the results including only breast cancer susceptibility genes can be created if necessary. 

However, a focus of this manuscript was to not only assess genes that were associated with breast cancer, but those that are available through commercially targeted cancer sequencing panels. Genes such as HOXB13 and FH are available on these multi-cancer panels. This was due to a concern that the breast cancer panels may have been developed and tested without the inclusion of a diverse population cohort. In fact, even while looking at these broad multi-cancer panels, there is a potentially limited utility for diverse populations such as for the African American women included here. 

As presented in the results section, genes such as HOXB13 and FH have been associated with other cancers. However, these populations were either of European descent or not reported. For example, the HOXB13 p.G84E variant was found not to be associated with breast cancer in women with European ancestry and was not detected in African or Asian populations (PMID: 32546843). However, the authors also acknowledge the sample size for individuals with African ancestry was limiting. Finally, during review of this manuscript, the African ancestry-specific occurrence of the X285K variant in HOXB13 was found to be associated with aggressive prostate cancer at an early age (PMID: 34799695).

The exome sequence data for these 85 genes should be made available (e.g. deposited into dbGAP). 

Exome data has been submitted to dbGAP.

Minor:

Gene names should be italicized.

All gene names are now italicized.

The term “mutation” (and deleterious mutation) should be replaced with “pathogenic variant”.

The term “mutation” was replaced with “variant” throughout the manuscript.

Line 70. Would read better as “the role of BRCA1/2 in…”

This has been changed as suggested as follows: “The role of BRCA1/2 in therapeutic regimens has emerged based on the DNA repair of double stranded breaks”

Lines 176-178. The sentence needs editing. “This variant is located in the forkhead-associated (FHA) functional domain of CHEK2 177 and while it has been found to impair DNA repair activity in yeast [38, 41].

This has been edited as follows: “This variant is located in the forkhead-associated (FHA) functional domain of CHEK2 and was found to impair DNA repair activity in yeast [38, 41].”

Reviewer #2: Disease history (other cancers, age, bilateral, and etc) of probands and cancer types of family members would be presented for explanation of low frequency of pathogenic varinat in this study group. Also it would be need to represent comparable study results for frequencies of PV, VUS, and benign variants

Similar points were also raised by reviewer 1. We have now modified the manuscript to include the following definition of family history and the inclusion criteria for high-risk individuals:

“Eligibility for this study included meeting one or more of the following criteria: 1) having multiple cases (≥2) of breast or ovarian cancer in the family; 2) having early onset breast cancer (≤40 years old); 3) having bilateral breast cancer; 4) having breast and ovarian cancer in the same individual, or 5) having male breast cancer in the family. Positive family history was defined as having multiple cases (≥2) of breast or ovarian cancer in the same side of the family.” The mean age for the probands, unaffected family members and affected family members were 52.3±15.5, 47.80±16.7, and 41.1±9.1, respectively. And, eleven cases reported personal histories of other cancers including, but not limited to lung, colorectal, and uterine cancer. 

Supplemental table 2 demonstrates frequencies in our population, 1000 Genomes Project, NHLBI GO Exome Sequencing Project (ESP), and Genome Aggregation Database (gnomAD).

---

## [Decision Letter · Decision Letter 1]

9 Aug 2022

PONE-D-22-08035R1Hereditary variants of unknown significance in African American women with breast cancerPLOS ONE

Dear Dr. Ricks-Santi,

Thank you for submitting your manuscript to PLOS ONE. After careful consideration, we feel that it has merit but does not fully meet PLOS ONE’s publication criteria as it currently stands. Therefore, we invite you to submit a revised version of the manuscript that addresses the points raised during the review process.

We look forward to receiving your revised manuscript.

Kind regards,

Alvaro Galli

Academic Editor

PLOS ONE

Journal Requirements:

Reviewers' comments:

Reviewer's Responses to Questions

**Comments to the Author**

1. If the authors have adequately addressed your comments raised in a previous round of review and you feel that this manuscript is now acceptable for publication, you may indicate that here to bypass the “Comments to the Author” section, enter your conflict of interest statement in the “Confidential to Editor” section, and submit your "Accept" recommendation.

Reviewer #1: (No Response)

2. Is the manuscript technically sound, and do the data support the conclusions?

Reviewer #1: Yes

3. Has the statistical analysis been performed appropriately and rigorously? 

Reviewer #1: Yes

4. Have the authors made all data underlying the findings in their manuscript fully available?

Reviewer #1: Yes

5. Is the manuscript presented in an intelligible fashion and written in standard English?

Reviewer #1: Yes

6. Review Comments to the Author

Reviewer #1: The authors have addressed all of my previous concerns. There are only 2 minor points to address.

1. In Table 1, the authors should replace SNP with missense. SNP suggests that the change is a benign polymorphism whereas missense more accurately describes the type of change.

2. the dbGAP accession number should be included in the manuscript.

7. PLOS authors have the option to publish the peer review history of their article (what does this mean?). If published, this will include your full peer review and any attached files.

Reviewer #1: No

---

## [Author Response · Author response to Decision Letter 1]

12 Aug 2022

We would like to thank the reviewers for their comments. Below, we have responded (in red) to each of the concerns raised. 

Reviewer #1: The authors have addressed all of my previous concerns. There are only 2 minor points to address.

1. In Table 1, the authors should replace SNP with missense. SNP suggests that the change is a benign polymorphism whereas missense more accurately describes the type of change.

The change has been made.

2. the dbGAP accession number should be included in the manuscript.

The accession number (phs002977) was added at line 112 of the document with no track changes.

---

## [Decision Letter · Decision Letter 2]

17 Aug 2022

Hereditary variants of unknown significance in African American women with breast cancer

PONE-D-22-08035R2

Dear Dr. Ricks-Santi,

We’re pleased to inform you that your manuscript has been judged scientifically suitable for publication and will be formally accepted for publication once it meets all outstanding technical requirements.

Kind regards,

Alvaro Galli

Academic Editor

PLOS ONE

Additional Editor Comments (optional):

Reviewers' comments:

Reviewer's Responses to Questions

**Comments to the Author**

1. If the authors have adequately addressed your comments raised in a previous round of review and you feel that this manuscript is now acceptable for publication, you may indicate that here to bypass the “Comments to the Author” section, enter your conflict of interest statement in the “Confidential to Editor” section, and submit your "Accept" recommendation.

Reviewer #1: All comments have been addressed

2. Is the manuscript technically sound, and do the data support the conclusions?

Reviewer #1: Yes

3. Has the statistical analysis been performed appropriately and rigorously? 

Reviewer #1: Yes

4. Have the authors made all data underlying the findings in their manuscript fully available?

Reviewer #1: Yes

5. Is the manuscript presented in an intelligible fashion and written in standard English?

Reviewer #1: Yes

6. Review Comments to the Author

Reviewer #1: (No Response)

7. PLOS authors have the option to publish the peer review history of their article (what does this mean?). If published, this will include your full peer review and any attached files.

Reviewer #1: No

---

## [Editor Report · Acceptance letter]

21 Oct 2022

PONE-D-22-08035R2 

Hereditary variants of unknown significance in African American women with breast cancer 

Dear Dr. Ricks-Santi:

I'm pleased to inform you that your manuscript has been deemed suitable for publication in PLOS ONE. Congratulations! Your manuscript is now with our production department. 

Kind regards, 

on behalf of

Dr. Alvaro Galli 

Academic Editor

PLOS ONE